# Epilepsy and Language Development in 8–36-Month-Old Toddlers with Tuberous Sclerosis Complex

**DOI:** 10.3390/jcm11154564

**Published:** 2022-08-04

**Authors:** Małgorzata Foryś-Basiejko, Katarzyna Kotulska, Agnieszka Maryniak, Agata Siłuszyk, Monika Szkop, Julita Borkowska, Monika Sugalska, Jagoda Głowacka-Walas, Sergiusz Jóźwiak

**Affiliations:** 1Department of Child Clinical Psychology and Family, Faculty of Psychology, University of Warsaw, 00-183 Warszawa, Poland; 2Department of Neurology and Epileptology, The Children’s Memorial Health Institute, 04-736 Warszawa, Poland; 3Department of Pediatric Neurology, Medical University of Warsaw, 02-091 Warszawa, Poland; 4Institute of Computer Science, Warsaw University of Technology, 00-665 Warszawa, Poland; 5Transition Technologies Science, 01-030 Warsaw, Poland

**Keywords:** language development, tuberous sclerosis complex, epilepsy, vocabulary, gesture production

## Abstract

This paper aimed to assess language development in infants and toddlers with tuberous sclerosis complex (TSC) and epilepsy, which increase the risk of autism spectrum disorder. We assessed language development in 61 patients with TSC at 8–36 months using a standardized Speech Development and Communication Inventory tool. The results showed differences in outcomes due to the duration of the seizures and the number of drugs (pFDR = 0.007 **—pFDR = 0.037 *). Children with TSC with longer epilepsy duration and receiving more antiepileptic drugs have a greater risk of language development delay.

## 1. Introduction

Tuberous sclerosis complex (TSC) is a multiorgan genetic disease inherited in an autosomal dominant way through a mutation of the *TSC1* or *TSC2* gene [1,2]. Patients with mutations in the *TSC2* gene are characterized by a more severe course of the disease. TSC is characterized by frequent comorbidity with epileptic seizures (72.5–90%) [3,4], intellectual disability (44%) [5], autism spectrum disorder (ASD) (17–61%) [6,7], and attention disorders (18–30%) [7,8]. Although delays in language development are often observed in patients with TSC, they have been rarely studied. Nevertheless, language development is an important topic to explore due to its impact on social and cognitive development and educational success. Early language development also significantly influences the quality of life of the entire family. The present paper aims to fulfill this gap. 

TSC significantly influences the development of children. The development of young patients with TSC depends on many factors, including the type of genetic mutation [4], history of epileptic seizures (e.g., age, type of seizures, response to pharmacotherapy) [4,9], and available therapies. TSC influences all areas of development, including cognitive, motor, and socio-emotional development. A study in children with TSC by Winterkorn et al. [10] found that a lower intelligence quotient was associated with a greater frequency of epileptic seizures, an earlier seizure onset, the occurrence of infantile spasms, *TSC2* gene mutation, and a family history of the disease.

Social development disorders are often reported in children with TSC, particularly ASD. The incidence of ASD within TSC is estimated to be 17–61%. A study by Jeste et al. [11] with 44 toddlers (19–36-month-olds) with TSC showed differences in social communication (e.g., the use of gestures, pointing, eye contact, social smile) between toddlers with and without ASD. There was a difference between toddlers with TSC and no ASD and healthy, typically developing toddlers as well [11].

In that report, four development areas were investigated: receptive and expressive language, visual reception, and fine motor. There were no specific profiles of scores associated with groups [11]. The results showed significant differences between the groups in the overall scores: the group of typically developing (TD) toddlers had the highest scores compared to the other groups. Importantly, the TSC/no ASD group obtained higher results than the TSC/ASD and nsASD groups (non-syndromic ASD—ASD without other comorbidities). The authors did not notice any significant difference between the TSC/ASD group and the nsASD group. Another study by Jeste and collaborators [12] compared various areas of development between healthy children and patients with TSC at 6, 12, and 18 months of age. The results indicate differences in expressive and receptive language appearing in the 12th month and lasting until the 18th month.

Studies by Jeste et al. suggest significant effects of TSC on development, even in the absence of neuropsychiatric comorbidities. The aforementioned results also indicate the occurrence of a continuum of development difficulties, including language and communication difficulties, which is consistent with various studies on the language development of children with ASD or at risk of ASD [13,14]. Children who struggle only with delayed language development are diagnosed with developmental language disorders (DLD). DLD occurs when the development in other areas is normal (e.g., cognitive, emotional, social). However, DLD may secondarily cause difficulties also in other spheres of a child’s development. Therefore, it is important to study language development in children with TSC [15].

Epilepsy affects 70–90% of patients with TSC, and in most cases, the first seizure appears very early in life [16,17]. Several studies suggest that 60–80% of patients with TSC experience their first epileptic seizures within the first three years of life, which is considered to be a period of exceptional neurodevelopmental sensitivity [7,17,18,19]. Early-onset epileptic seizures have been shown to increase the risk of neurodevelopmental and cognitive disorders [20,21,22]. The mean of the standardized scores of longitudinal research by Grayson et al. [23] pointed to typical psychomotor, language, and adaptive development in 3–36-month-old toddlers with TSC without epilepsy. Standardized test scores were better in toddlers with TSC (but not epilepsy, e.g., mean: 98–101) as compared to those with pharmacologically controlled epilepsy (mean: 87–88 standard score). Importantly, toddlers with drug-resistant epilepsy had the lowest results (mean: 77–83 standard score). However, it is important to note that the results in all groups had high degrees of variability. These studies showed the considerable influence of epilepsy and effective treatments on the development of children with TSC. A detailed analysis of the raw data regarding expressive and receptive language among children with TSC revealed that scores were poorer in children with epilepsy compared to those without epilepsy. There was also an age-related increase in the raw scores. Although these results suggest a slower rate of learning in children with concurrent epilepsy and TSC, they also highlight the developmental potential of this group [23]. Of note, study groups (children with TSC and ASD vs. children with TSC but no ASD) in the aforementioned report by Jeste et al. [11] were comparable regarding various aspects of epilepsy, including age, seizure onset, seizure history, number of drugs, and recent frequency of seizures. However, in another Jeste et al. study [12], the study groups (TSC and ASD vs. TSC without ASD) differed statistically significantly in total duration of seizure and the number of used antiepileptic drugs. 

The present study aimed to examine the impact of epilepsy on language development in toddlers with TSC. We examined speech and communication development and tested the impact of epilepsy on expressive language (i.e., production), receptive language (i.e., comprehension), and non-verbal communication (i.e., gestures). To our knowledge, there are a few studies that examine language development and epilepsy in young children with TSC; however, none of them consider gestures as a measure of language development. The specifics of our study also include the inclusion of young TSC patients without clinical seizures but who already had EEG seizure changes, which was included in the descriptive data.

## 2. Materials and Methods

### 2.1. Study Participants

Sixty-one toddlers with TSC participated in this study (28 females, 33 males). All of the participants received treatment between the years 2017 and 2019 in the Pediatric Neurology Department of the Medical University of Warsaw or in the Neurology and Epileptology Department of the Children’s Memorial Health Institute in Warsaw. 

All of the patients were diagnosed with TSC according to the criteria of the International TSC Consensus Conference from 2012 [24]. The study included patients treated conservatively, that is, started after the occurrence of clinical seizures (*n* = 27), patients treated preventively with antiepileptic drugs after observing epileptic discharges in EEG but before clinical seizures (*n* = 24) (i.e., an abnormal EEG, abEEG), and patients who did not receive antiepileptic drugs due to lack of abEEG or clinical seizures (*n* = 9). None of the participants underwent epilepsy surgery.

To study how many of our participants are at high risk of delayed language development. We applied a border point to score 10 percentiles according to the Manuals of SCDI [25]. Table 1 presents detailed characteristics of the studied groups, including information about age and genetic alteration.

### 2.2. Research Tools

#### 2.2.1. The Speech Development and Communication Inventory

The Speech Development and Communication Inventory (SDCI) is the Polish version of the Communication Disorder Inventory (CDI) developed by MacArthur-Bates. The SDCI is a standardized parent-report assessment tool allowing percentile evaluation of expressive vocabulary in 8–36-month-old toddlers and receptive vocabulary using gestures and simple play in 8–18-month-old toddlers. Due to the dynamic development associated with the toddler period, there are two versions of the SDCI: (1) Words and Gestures and (2) Words and Sentences. 

SDCI Words and Gestures (SCDI-WG) is dedicated to children aged 8–18 months. It contains a list of 380 words. Parents are instructed to determine whether the words will be understood and used by the child. The list of words is divided into 19 categories, such as food or activities. The second part of the questionnaire comprises 63 gestures and activities performed by the child, divided into five categories (e.g., caring). 

SDCI Words and Sentences (SCDI-WS) is dedicated to toddlers aged 18–36 months. It contains a list of 670 words and encompasses only expressive vocabulary. The SDCI Words and Sentences also includes questions regarding the linking of words.

There are two percentile scales for both versions allowing the interpretation of raw results: (1) according to sex and (2) a general scale. In this study, we used the general scale that does not differentiate the results according to sex. The percentile norms are for each month in the range of 8–36 months. The SCDI is a reliable test for measuring the linguistic development of young children. Standardization studies were conducted on a group of 3331 children. Cronbach’s alpha, adequate to the studied groups, was 0.87–0.99. The time stability tested by correlation rho Spearman ranged from r = 0.85 to r = 0.97 [25]. Using standardized tools and norms provides a reference to an objective group of healthy and thus to a control group. 

According to Sachse and Von Suchodoletz [26], a parent-report questionnaire is a credible form of language development assessment among children in infancy and toddler age. They showed a correlation between the parent-report questionnaire and a direct language assessment: a strong correlation in vocabulary (word)—production and a weaker correlation in vocabulary (word)—comprehension. The literature shows that SCDI correlates satisfactorily with other measures of development in the groups of children with developmental delayed or autism spectrum disorders as well [27,28,29]. 

The infant questionnaire (SDCI Words and Gesture) was used on 34 participants (8–18-month-olds, mean: 12.5 months). The young child questionnaire (SDCI Words and Sentences) was used on 27 participants (19–36-month-olds, mean: 25 months). 

#### 2.2.2. Questionnaire Concerning Epilepsy

We developed a parent-report questionnaire regarding the development of epilepsy to assess the association between epilepsy and speech development. The questionnaire was based on the modified Early Childhood Epilepsy Severity Scale (E-CHESS) [30]. The original questionnaire consists of three questions (on a 1–3 scale) from the E-CHESS questionnaire, pertaining to (1) the period of seizure occurrence (1 = less than a month; 2 = 1–6 months; 3 = over 6 months); (2) number of used antiepileptic drugs (open-ended question, responses categorized according to the following scale: 1 = no drugs; 2 = 1 drug; 3 = more than 1 drug); (3) response to treatment (1 = full remission of seizures; 2 = reduced number of seizures; 3 = no change in the number of seizures). The questionnaire also includes questions regarding (1) the age of seizure onset (open-ended question, responses categorized according to the following scale: 0 = in 6 months of life or earlier; 1 = after 6 months of life); (2) reason for treatment (0 = abnormal EEG; 1 = occurrence of epileptic seizure); (3) the current occurrence of seizures (0 = no; 1 = yes). The questionnaire used in this study did not include two questions from the E-CHESS questionnaire regarding the frequency and seizure types. 

The original E-CHESS questionnaire classifies epilepsy as drug-resistant if two different drugs are not effective for reducing seizure frequency (scoring system: 1 = no drugs; 2 = 1 or 2 drugs; 3 = more than 2 drugs). In our study, we classified epilepsy as drug-resistant if one drug was deemed ineffective and thus another drug was added (1 = no drugs; 2 = 1 drug; 3 = more than 1 drug). Vigabatrin (dose: 100–150 mg/kg/day) was the first-line drug for all patients in this study. Thus, the results concerning the group taking at least two drugs refer to patients who have already used vigabatrin.

### 2.3. Statistical Analyses

To examine the relationship between epilepsy and language development, we compared the particular language skills between patients who differed in [1] terms of their course of epilepsy. The statistical analyses were conducted by using the percentile scale of SCDI. Two nonparametric tests were used—the Wilcoxon rank sum test with continuity correction (2 groups) and the Kruskal–Wallis test (3 groups). Comparative groups were selected based on a questionnaire filled in by parents: seizure onset (2 groups), the reason for starting treatment (2 groups), the occurrence of the seizure (2 groups), number of drugs (3 groups), and response to treatment (2 [8–36 months and 19–36 months] or 3 [8–18 months] groups). Post-hoc analyses were carried out using Wilcoxon’s signed-rank test with the Benjamini–Hochberg Procedure.

The statistical analyses were carried out using Statistica software and R statistical software. The results were considered significant at a *p* < 0.05 threshold. The results that reached *p* < 0.01 ** and *p* < 0.05 * were marked with asterisks for clarity.

## 3. Results

### 3.1. Language Development in Patients with Various Characteristics of Epilepsy

Table 2 presents the relationship between the particular aspects of speech development (expressive vocabulary, receptive vocabulary, and gestures) and various characteristics of epilepsy (age of seizure onset, reason for antiepileptic treatment introduction, duration of the seizure, number of currently taken antiepileptic drugs, and response to current treatment). There were statistically significant effects for expressive vocabulary for the entire studied group (H = 12.19, *p* = 0.002 **), and 19–36-month-olds (H = 6.36, *p* = 0.042 *) depending on the number of drugs. There were also significant effects for receptive vocabulary (H = 10.88, *p* = 0.004 **) and gestures (H = 9.54, *p* = 0.009 **) in 8–18-month-olds depending on the number of drugs as well. There were also differences in expressive vocabulary across the entire study sample regarding the duration of seizures (H = 13.22, *p* = 0.001 **) and response to treatment (H = 6.14,88, *p* = 0.047 *) (Table 2). All quartile plots are enclosed in the Appendix A.

### 3.2. Post-Hoc Analyses

Table 3 presents results of a more detailed analysis of the expressive vocabulary results across the entire study sample (8–36-month-olds). 

The data suggest that children with seizures lasting 6 months or less (for less than 1 month *Me* = 40, for 1–6 months group *Me* = 25) achieve better results than children with seizures lasting more than 6 months (*Me* = 5) (*p_FDR_* = 0.0089 ** between seizures lasting less than a month and more than 6 months and *p_FDR_* = 0.0014 ** between seizures lasting 1–6 months and more than 6 months). A detailed distribution of the results is shown in Figure 1.

Subsequently, we performed a more detailed analysis of the results for each particular SDCI subscale. We examined the dependence of a number of drugs on expressive vocabulary, receptive vocabulary, and gestures. We found that participants who took one drug obtained better scores compared to participants who took at least two drugs. In expressive vocabulary across the entire study sample group, the at least two-drugs group (*Me* = 5) had a lower score than the no-drugs group (*Me* = 40, *p_FDR_* = 0.007 **) and the one-drug group (*Me* = 25, *p_FDR_* = 0.010 *). In expressive vocabulary across the older sample group (19–36 months), at least two drugs (*Me* = 1) had a lower score than the no-drugs group (*Me* = 30, *p_FDR_* = 0.037 *). In the younger group (8–18 months) in receptive vocabulary, the at least two-drugs group (*Me* = 1) had a lower score than the no-drugs group (*Me* = 60, *p_FDR_* = 0.020 *) and one-drug group (*Me* = 17.5, *p_FDR_* = 0.015 *). Post-hoc analysis of scores of gestures shows similar pattern: at least two-drugs group (*Me* = 1) had a lower score than the no-drugs group (*Me* = 45, *p_FDR_* = 0.025 *) and one-drug group (*Me* = 5, *p_FDR_* = 0.036 *). 

## 4. Discussion

### 4.1. The Role of Gestures in Speech Development

In our study, we wanted to find out what are the relationships between the severity of epilepsy and the development of speech. The results concerning the development of gestures are particularly interesting, as it has not been the subject of previous research. We examined how various aspects of epilepsy influence using gestures and expressive and receptive vocabulary. Most of all, our results indicate differences between the group with two antiepileptics and one antiepileptic or no drugs in all studied language areas. There are also significant differences between children with seizures lasting more than 6 months and groups with seizures lasting less than one month and lasting 1–6 months as well. Additionally, Table 1 shows that in our group of all examined patients, 26% were high-risk delayed language development patients (HR-DLD) considering gestures, 53% HR-DLD considering receptive vocabulary, and 37% HR-DLD considering expressive vocabulary in the 19–36 months group. 

Our results on gestures and comprehension in the younger group indicate a relatively high percentage of children at risk of delayed language development and, as the research shows, the risk of being diagnosed with an autism spectrum disorder. Botting et al. [31] compared the role of gestures in typically developing children (TD) and in children with specific language impairment (SLI, now known as DLD—developmental language disorder) during the pre-school and early school age. In that study, children with SLI tended to base their communication on gestures, whereas TD children stopped using gestures as they grew older [31]. These results suggest that gestures are an important communication tool for children with language disorders. Studies on other communication disorders also suggest the vital role of gestures in early speech development [32,33,34].

*Gestures* subtest in questionnaire SCDI-WG, applied in our study, evaluates the use of simple gestures (e.g., pointing, clapping) and simple imitation play (e.g., caring activities). The lack of such activities until 18 months of age is interpreted as a developmental deficit and is a prerequisite for performing diagnostic assessments for ASD [35,36]. Mitchell et al. [14] used the CDI questionnaire in their study to compare the language development of ASD-diagnosed, ASD-sibling without diagnosis (noASD), and control infants in the 12th and 18th months. Their results show significance in gestures between the ASD group and both noASD and control in the 12th and 18th month. West et al. [37] and Choi et al. [38] also write about the importance and supportive role of gestures in the early detection of autism: their research showed that the ASD group in infancy used fewer gestures compared to the control group. Jeste et al. [11] have suggested that the negative effects of a TSC diagnosis on language development in the absence of ASD may be interpreted in one of two ways. On the one hand, deficits in some social skills may be directly associated with TSC without comorbid ASD [11]. On the other hand, there may also be a group of patients with TSC who have deficits in social communication but no deficits in rigid behavior patterns. In the dimensional diagnosis, these patients would be considered to be within the normal range of possible scores. In the study of Jeste et al. [11], children in the TSC (no ASD) group differed from TD children in terms of ASD symptoms presented as ADOS-2 results. The analyses showed a difference in the mean ADOS-2 result but not in the pattern.

In our study, we did not find any differences in expressive vocabulary between children with TSC with and without a history of clinical seizures. On the one hand, this null finding may suggest that the occurrence of epilepsy does not influence the results of the language development evaluation. On the other hand, the lack of findings may be caused by a lack of power in the study or an unrepresentative study group.

### 4.2. The Influence of Epilepsy on Language Development

Differences in language development in children with different characteristics of epilepsy are notable in our study group. Similar results were reported by Schoenberger and colleagues [39] in a study on the impact of epilepsy on speech development and ASD in 12–36-month-olds. In that study, the occurrence of epileptic seizures before 6 months of age was associated with an ASD diagnosis at 36 months of age. Further, more frequent seizures were correlated with poorer language development scores. Another study by Jurkevičiene et al. [40] in a group of older children found that the younger age of epilepsy onset predicted an increased risk of language disorders. Therefore, the results of our study, together with previous reports, suggest the importance of early and effective treatment or prevention of epilepsy to improve language development.

In our study, we also observed the relationship between the number of drugs and language development. In our study group, a second drug was introduced to patients who showed the occurrence or increase in the frequency of epileptic seizures despite treatment with vigabatrin. Currently, vigabatrin is commonly used as the first-choice drug to treat seizures in infants and young children with TSC. Vigabatrin is also used to prevent epileptic seizures in infants with TSC [17,41]. Therefore, in the present study, the ineffectiveness of vigabatrin to prevent epilepsy or to achieve seizure remission after the occurrence of epileptic seizures was used as a marker of epilepsy severity. Children taking at least two drugs had poorer results in almost all studied areas, including general production, compared to children who did not require antiepileptic treatment (across the entire study sample of 8–36-month-olds and in older children [i.e., 19–36-month-olds], in particular), comprehension (in 8–18-month-olds), and gestures (in 8–18-month-olds). The change of the “cut-off” point for drug-resistant epilepsy from a minimum of three to a minimum of two drugs correctly differentiated the groups. This change also highlighted the differences between patients and, as a result, allowed the better characterization of patients in the present study.

In this context, it is important to consider the differences between children who took two antiepileptic drugs and children who took one drug. Children with one drug showed better results in receptive vocabulary and gestures (8–18-month-olds in both) as compared to children in whom the second drug was introduced. This result is crucial because of the important role of speech comprehension and gestures in young children for scaffolding future language development. On the other hand, these patterns were not observed in older children concerning their speech production. Our results on the number of AEDs are consistent with the findings of Kadish et al. [42]. For children who were off medication and for children who were taking one medication, no difference was seen in the areas studied. This is valuable knowledge regarding the pharmacotherapy of AEDs.

Our data highlight the possible critical impact of the duration of seizures and the effects of treatment on expressive vocabulary. Children with seizures lasting more than 6 months achieved poorer results than children with a shorter seizure duration. It shows that achieving quick seizure remission may be pivotal for speech development in children with TSC. Therefore, antiepileptic treatment should be quickly implemented after the first seizure to increase the chance of seizure remission. Taking into account recently published results of preventive antiepileptic treatment with vigabatrin in TSC, such an approach should also be considered [17]. Analogically, children who experienced full seizure remission had better results compared to children whose treatment was ineffective. Similar results were obtained in the study of Wu et al. [43], wherein uncontrolled epilepsy influenced general development at the age of two years. The study of Wu et al. [43] also demonstrates the importance of early monitoring of epilepsy risk for speech development in young children with TSC. These data also highlight the crucial role of early diagnosis of TSC. A retrospective study by Cusmai and colleagues [44] and a prospective study by Jóźwiak [41] and colleagues found that children with earlier-introduced antiepileptic treatment had better developmental results than children treated later [19,44,45]. These benefits were reflected in the less frequent diagnosis of intellectual disability and ASD and more frequent remission of seizures. Currently, the advances in modern research techniques allow for the diagnosis of TSC in early infancy or even prenatally [46]. According to the recommendations of the European group of TSC experts, prenatal diagnosis of TSC will enable clinicians to monitor the risk of epilepsy by performing regular EEG studies in newborns and infants [47,48]. Prenatal diagnosis also allows the introduction of preventive treatment when epileptic alterations are detected on EEG before the onset of clinical seizures [47,48]. Future research should use prospective designs to examine the effectiveness of early intervention in delayed language development. 

There are limitations of this study that need to be underlined. Firstly, there were relatively limited sample sizes in certain subgroup analyses. Secondly, some aspects of epilepsy were evaluated subjectively by parents, and thus we lacked an objective measure (e.g., the effectiveness of treatment measured with remission or decrease in the number of seizures). Thirdly, we examined linguistic development only at a one-time point. Speech and gesture comprehension in older children (between 19–36 months of age) is an important avenue for further research, including longitudinal research.

## 5. Conclusions

In conclusion, the presented study contributes to the understanding of language and communicative difficulties in children with TSC. The results constitute a detailed analysis of the characteristics of speech development in infants with TSC who are at high risk of ASD and intellectual disabilities comorbidity. We also present the clinical implications of the effects of epilepsy on language development. Given the limited number of studies in this area, future research is needed to broaden the knowledge about the impact of epilepsy on early language development. In view of the above considerations, conducting research that takes the early development of gestures and non-verbal communication into account seems to be of particular importance. A valuable observation for clinical practice is that when assessing the linguistic development of young patients, attend not only to the vocabulary but also communicate with gestures.

## Figures and Tables

**Figure 1 jcm-11-04564-f001:**
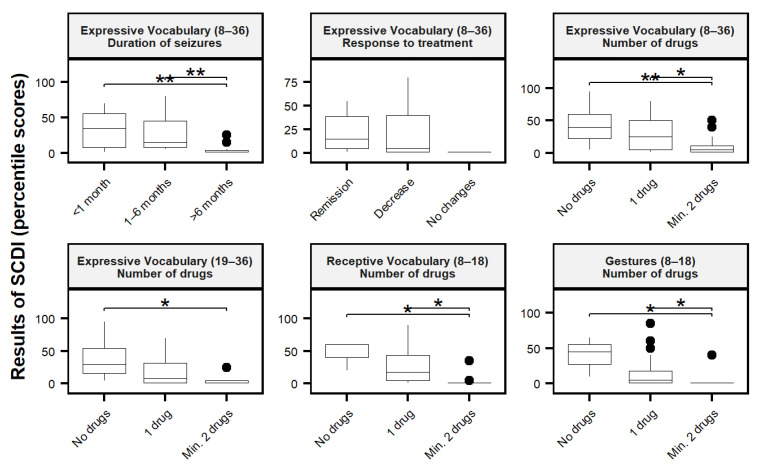
Quartile distribution of statistically significant results: duration of seizures, response to treatment, and the number of drugs. The Y axis indicates the SCDI percentile score. On the X axis, there are names of groups of individual variables. * *p* < 0.05, ** *p* < 0.01 range.

**Table 1 jcm-11-04564-t001:** Descriptive statistics for age of the study participants and number of study participants with regard to sex, the reason for treatment introduction, genetic alteration, age of the first occurrence of seizures and risk of delayed language development.

		8–18 Months (SCDI-WG)	19–36 Months (SDCI-WS)	All
		*n* = 34	*n* = 27	*n* = 61
**Age (in months)**				
Mean		12.7	26.1	18.7
SD		2.4	4.8	7.6
median		12.5	25	16
quartile 1		12	23	12
quartile 3		14	28.5	24
**Sex**				
Female		16 (47%)	12 (44%)	28 (46%)
Male		18 (53%)	15 (56%)	33 (54%)
**Reason for treatment**				
No treatment needed		3 (9%)	5 (19%)	8 (13%)
abEEG		15 (44%)	10 (37%)	25 (42%)
Clinical seizures		16 (47%)	11 (41%)	27 (44%)
No information		NA	1 (3%)	1 (2%)
**Identified genetic alteration**				
*TSC1*		3 (9%)	2 (7%)	5 (8%)
*TSC2*		14 (41%)	4 (15%)	18 (30%)
No identify		2 (6%)	0	2 (3%)
No information		15 (44%)	21 (78%)	36 (59%)
**Age at first seizure**				
No seizures		11 (32%)	9 (33%)	20 (33%)
First seizure: 1–12 months		21 (62%)	11 (41%)	32 (52%)
First seizure: 13–24 months		1 (3%)	6 (22%)	7 (12%)
No information about first seizure		1 (3%)	1 (4%)	2 (3%)
**Expressive Vocabulary**				
≤10 percentile (HR-DLD)		23 (77%)	10 (37%)	33 (54%)
>10 percentile		11 (32%)	17 (53%)	28 (46%)
**Receptive Vocabulary**				
≤10 percentile (HR-DLD)		18 (53%)	NA	NA
>10 percentile		16 (47%)	NA	NA
**Gestures**				
≤10 percentile (HR-DLD)		9 (26%)	NA	NA
>10 percentile		25 (74%)	NA	NA

SCDI-WG—Speech Development and Communication Inventory-Word and Gestures, SCDI-WS—Speech Development and Communication Inventory-Word and Sentences, SD—standard deviation, abEEG—epileptic alterations on EEG, HR-DLD—High risk of delayed language development.

**Table 2 jcm-11-04564-t002:** The comparison of results of the SDCI questionnaire according to age of seizure onset, the reason for antiepileptic treatment introduction, duration of the seizure presence, number of currently taken antiepileptic drugs, and observed response to present treatment (i.e., decrease in seizure, seizure remission, or no response).

		8–36 Months	19–36 Months	8–18 Months		
		Expressive Vocabulary	Expressive Vocabulary	Expressive Vocabulary	Receptive Vocabulary	Gestures
**Epilepsy onset (age in months)**	*n*	39	17	22	22	22
	*p*	0.064	0.094	0.167	0.15	0.614
	*p_FDR_*	0.209	0.209	0.209	0.209	0.614
	*U*	122.5	20	37.5	37.5	51.5
**Reason for treatment**	*n*	52	21	31	31	31
	*p*	0.42	0.796	0.126	0.174	0.083
	*p_FDR_*	0.557	0.892	0.290	0.290	0.290
	*U*	393.5	63.5	159	154	161
**Occurrence of seizures**	*n*	41	18	23	23	23
	*p*	0.531	0.879	1	0.888	0.438
	*p_FDR_*	1	1	1	1	1
	*U*	172	38	56.5	53.5	45
**Duration of seizures**	*n*	36	17	19	19	19
	*p*	0.001 **	0.149	0.145	0.109	0.193
	*p_FDR_*	0.005 *	0.186	0.186	0.186	0.193
	*H*	13.218	3.81	3.86	4.43	3.29
**Number of drugs**	*n*	60	27	34	34	34
	*p*	0.002 **	0.042 *	0.054	0.004 **	0.009 **
	*p_FDR_*	0.010 *	0.053	0.054	0.010 *	0.013 *
	*H*	12.19	6.36	5.82	10.88	9.54
**Response to treatment**	*n*	36	16	20	20	20
	*p*	0.047 *	0.192	0.803	0.828	0.836
	*p_FDR_*	0.480	0.230	0.828	0.828	0.828
	*H*/*U*	6.14 *^H^*	3.30 *^H^*	45.5 *^U^*	39 *^U^*	45 *^U^*

*n*—number of study participants in the analysis; *p*—statistical significance; *p_FDR_*—statistical significance after continuity correction; *U*—Wilcoxon test statistic; *H*—Kruskal–Wallis statistic. * *p* < 0.05, ** *p* < 0.01.

**Table 3 jcm-11-04564-t003:** Results of post-hoc analyses for the tested variables in expressive vocabulary, receptive vocabulary, and gestures.

Duration of Seizures				Response to Treatment			
	Expressive Vocabulary (8–36)				Expressive Vocabulary (8–36)		
	<1 month	1–6 months	>6 months		Seizure remission	Decrease in number of seizures	No changes
	(*n* = 11, *Me* = 35)	(*n* = 11, *Me* = 15)	(*n* = 14, *Me* = 1)		(*n* = 20, *Me* = 15)	(*n* = 13, *Me* = 5)	(*n* = 3, *Me* = 1)
<1 month	x	*p_FDR_* = 0.791	*p_FDR_* = 0.0089 **	Remission	x	*p_FDR_* = 0.210	*p_FDR_* = 0.062
1–6 months		x	*p_FDR_* = 0.0014 **	Decrease		x	*p_FDR_* = 0.205
>6 months			x	No changes			x
**Number of drugs**							
	Expressive Vocabulary (8–36)				Expressive vocabulary (19–36)		
	No drugs	1 drug	min. 2 drugs		No drugs	1 drug	min. 2 drugs
	(*n* = 7, *Me* = 40)	(*n* = 33, *Me* = 25)	(*n* = 20, *Me* = 5)		(*n* = 4, *Me* = 30)	(*n* = 12, *Me* = 8)	(*n* = 11, *Me* = 1)
No drugs	x	*p_FDR_* = 0.157	*p_FDR_* = 0.007 **		x	*p_FDR_* = 0.196	*p_FDR_* = 0.037 *
1 drug		x	*p_FDR_* = 0.010 *			x	*p_FDR_* = 0.196
min. 2 drugs			x				x
	Receptive Vocabulary (8–18)				Gestures (8–18)		
	No drugs	1 drug	min. 2 drugs		No drugs	1 drug	min. 2 drugs
	(*n* = 3, *Me* = 60)	(*n* = 22, *Me* = 17.5)	(*n* = 9, *Me* = 1)		(*n* = 3, *Me* = 45)	(*n* = 22, *Me* = 5)	(*n* = 9, *Me* = 1)
No drugs	x	*p_FDR_* = 0.152	*p_FDR_* = 0.020 *		x	*p_FDR_* = 0.065	*p_FDR_* = 0.025 *
1 drug		x	*p_FDR_* = 0.015 *			x	*p_FDR_* = 0.036 *
min. 2 drugs			x				x

*n*—number of study participants in the analysis; *Me*—Median; *p_FDR_*—statistical significance with Benjamini-Hochberg Procedure. * *p* < 0.05, ** *p* < 0.01.

## Data Availability

Not applicable.

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
