# Peer review of "Epilepsy and Language Development in 8–36-Month-Old Toddlers with Tuberous Sclerosis Complex"

_jcm, 2022, doi:10.3390/jcm11154564_

Round 1
Reviewer 1 Report
Please state in the Discussion section of the manuscript what is the novelty of your study? There is no control groups. Study would benefit from inclusion of healthy control groups of month-age matched children without neurological, cognitive, or epilepsy co-morbidites, and moght-age matched children with epilepsy but not secondary to TSC. The first sentence of the first paragraph of the 2.3. Statistical Analysis section does not belong there and should be moved to a special section dedicated to the Study design or Study participants. There is no follow-up so your study is not enabling you to conclude that early and effective treatment of epilepsy improves language development.
Reviewer 2 Report
-Any patient underwent epilepsy surgery has to be mentioned included the type of surgery
- the conclusion at the end written twice.
Reviewer 3 Report
Forys-Basiejko et al studied language development of toddlers with tuberous sclerosis complex, and found the duration of epileptic seizures and the number of antiepileptic drugs (or ineffectiveness of the first drug, vigabatrin) as risks of delay.
This study is well designed, and the results are nicely shown.
The reviewer sees some issues, and have criticisms too:
1. In the Introduction and Discussion. analyses about the mechanism of speech delay are confusing. A delay in language development may be caused by many conditions including intellectual disability (ID), autism spectrum disorder (ASD) and developmental language disorders (DLD), of which children with TSC may have one or more. Which is the most important for delay in language development?
2. In this study, it is of course impossible to make a solid diagnosis of these conditions at 8-36 months of age. This may be the biggest limitation of this study.
3. The abbreviation “HR-DLD” for “high-risk language development” is extremely confusing because DLD also stands for “developmental language disorders”.
Round 2
Reviewer 3 Report
Authors have duly addressed the issues in the original version.